# Flow parallel synthesizer for multiplex synthesis of aryl diazonium libraries via efficient parameter screening

Gwang-Noh Ahn [1,2], Brijesh M. Sharma[1,2], Santosh Lahore[1], Se-Jun Yim[1], Shinde Vidyacharan[1] & Dong-Pyo Kim [1✉]

The development of miniaturized flow platforms would enable efficient and selective synthesis of drug and lead molecules by rapidly exploring synthetic methodologies and screening for optimal conditions, progress in which could be transformative for the field. In spite of tremendous advances made in continuous flow technology, these reported flow platforms are not devised to conduct many different reactions simultaneously. Herein, we report a metal-based flow parallel synthesizer that enables multiplex synthesis of libraries of compounds and efficient screening of parameters. This miniaturized synthesizer, equipped with a unique built-in flow distributor and $n$ number of microreactors, can execute multiple types of reactions in parallel under diverse conditions, including photochemistry. Diazonium-based reactions are explored as a test case by distributing the reagent to 16 ($n = 16$) capillaries to which various building blocks are supplied for the chemistry library synthesis at the optimal conditions obtained by multiplex screening of 96 different reaction variables in reaction time, concentration, and product type. The proficiency of the flow parallel synthesizer is showcased by multiplex formation of various C–C, C–N, C–X, and C–S bonds, leading to optimization of 24 different aryl diazonium chemistries.

---

[1] Department of Chemical Engineering, Center for Intelligent Microprocess Pharmaceutical Synthesis, Pohang University of Science and Technology (POSTECH), Pohang 790-784, Republic of Korea. [2] These authors contributed equally: Gwang-Noh Ahn, Brijesh M. Sharma. ✉email: dpkim@postech.ac.kr

Continuous-flow technology[1–3] for chemical synthesis offers better reproducibility, higher selectivity and better control over various reaction parameters than batch technology[4–10]. The advent of the technology has led to the development of "universal" automated flow synthesis platform for efficient optimization of organic synthesis[11,12], computer-aided synthesis planning (CASP) and robotically executed target-oriented or diversity-oriented flow synthesis of specific molecules for exploration of new drug development[13–22]. In addition, on-demand synthesis of small molecules[18], use of reconfigurable system[23] and arrays of commercially available multiple reactor modules have elevated the flow capabilities to a new horizon, which are mostly focused on the synthesis of target molecules.

In general, to select a specific target molecule requires numerous screening synthesis tests and optimization. The traditional batch approach of "design–synthesis–screen" for the discovery of a lead molecule is time-consuming and/or at high risk with low probability of success[24]. Optimization and screening of variables for a chemical reaction is an issue that has troubled synthetic community throughout the history[25]. Currently, synthetic community mostly relies on commercially available batch synthesizers for quick screening of solvents, reaction temperatures, reagents and concentrations for high product yield and purity. These batch parallel synthesizers usually require simultaneous handling and integration of a large number of reaction flasks on a single platform. A screening approach based on microwell platform allowed a high throughput in identifying additive combinations for organic reactions[26–31]. A recent elegant approach of handling reagents in nano-liter volumes using microliter plate system in batch mode enhanced the efficiency of optimization of an early phase of any natural product or drug discovery programme[18,32]. Formation of the "A × B" libraries in flow are mostly based on segmented flow system with highly enhanced mass and heat transfer, leading to screen catalysts[33–35], combinatorial chemistry[12,17] and nanomaterials[36]. However, these approaches have limitations in direct utilization of the derived optimal reaction conditions into continuous-flow process.

On the other hand, in spite of the tremendous advances made for the continuous-flow technology for synthesizing a given target molecule, the reported flow platforms are mostly based on either linear or radial approach to perform single or multistage transformation in a sequential way, and they are not devised to conduct many different reactions simultaneously[22]. It is rather ironic that no significant progress has been made in utilizing continuous-flow synthesizer for the purpose of synthesis, screening and optimization, except a primitive $2 \times 2$ parallelized capillary system that enabled to conduct only several reactions[37]. From chemist's perspective, it would be highly desirable to have a flow parallel synthesizer that enables synthesis combinations of $A_i \times (B_1, B_2, B_3... B_n)$ in a multiplex mode in one single action, instead of multiple $A_i \times B_j$ type reactions sequentially as shown in Fig. 1.

Here, we present a metal-based flow parallel synthesizer for synthetic screening and optimization. This flow parallel synthesizer has a damper-based distributor with baffles that enables uniform flow distribution and allows the system to operate without interruption when clogging occurs in some of the capillaries. Therefore, the synthesizer can concurrently execute multiple reactions in parallel under diverse reaction conditions, including photochemistry. This desk-top type of synthesis platform is utilized to explore diazonium-based reactions as a test case for multiplex screening of 96 different reaction variables in reaction time and concentration in building the chemistry library. The proficiency of the flow parallel synthesizer is demonstrated by enabling multiplex formation of various C–C, C–N, C–X and C–S bonds, leading to optimization of 24 different aryl diazonium chemistries, including a scaling-up test, simply by switching aryl diazonium salts. No reconfiguration of modules and components is involved in demonstrating the capabilities of this flow parallel synthesizer that can expedite the workflow in the discovery stage for a lead molecule.

## Results and discussion

**Design principle of flow parallel synthesizer**. To date, parallel type of microreactors have been mainly concerned for the purpose of high-throughput production[38–44]. In this work, the concept of parallelization is used to develop the new approach for the reaction screening and optimization of chemistry. The metal-based flow parallel synthesizer in Fig. 2 (Supplementary Fig. 1, see Supplementary Information for the details) is an assembly of multiple modules with specific functions. These modules are a flow distributor embedded with a baffle disc, two inlets at the bottom of the distributor, a set of independent individual outlets connected to the same number of mixers and a set of coiled capillaries of microreactors[45].

One main design consideration was to ensure even flow distribution to the feed lines from the distributor in spite of

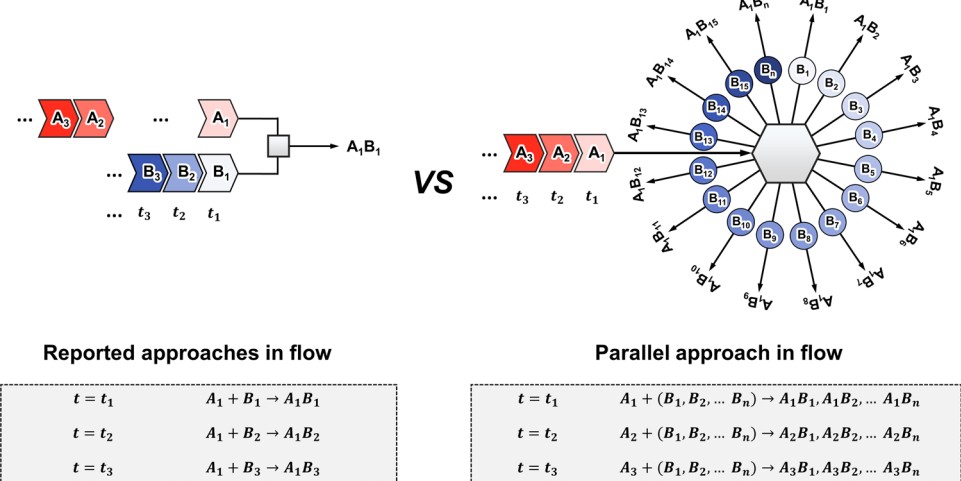

| Reported approaches in flow | Parallel approach in flow |
|---|---|

| $t = t_1$ | $A_1 + B_1 \rightarrow A_1B_1$ |
| $t = t_2$ | $A_1 + B_2 \rightarrow A_1B_2$ |
| $t = t_3$ | $A_1 + B_3 \rightarrow A_1B_3$ |

| $t = t_1$ | $A_1 + (B_1, B_2, ... B_n) \rightarrow A_1B_1, A_1B_2, ... A_1B_n$ |
| $t = t_2$ | $A_2 + (B_1, B_2, ... B_n) \rightarrow A_2B_1, A_2B_2, ... A_2B_n$ |
| $t = t_3$ | $A_3 + (B_1, B_2, ... B_n) \rightarrow A_3B_1, A_3B_2, ... A_3B_n$ |

**Fig. 1 Comparison of the reported approach and our parallel approach in flow system.** The reported approaches (left) in flow system mostly performs $A_i \times B_j$ type reactions sequentially. On the other hand, our parallel approach (right) uses the flow parallel synthesizer to enable composite combinations of $A_i \times (B_1, B_2, B_3... B_n)$ in multiple modes in a single operation.

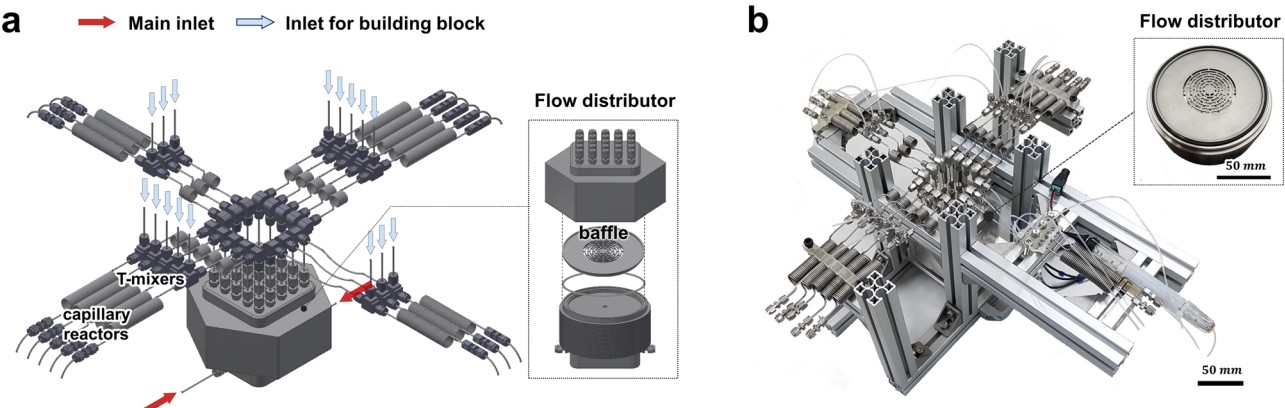

**Fig. 2 Flow parallel synthesizer for multiplex synthesis. a** Schematic diagram and **b** photographs of the flow parallel synthesizer assembled with 16 capillaries to which the reagent injected by two main inlets at the bottom is uniformly supplied in upward flow through a reservoir type of distributor, consecutively merged with diverse building blocks at each T-mixer.

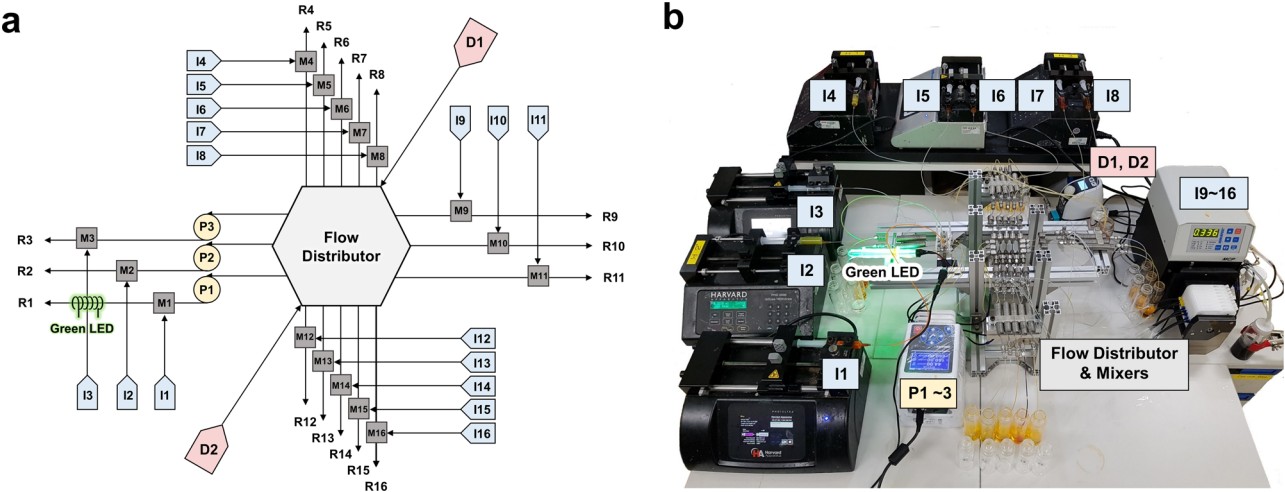

**Fig. 3 Parallel flow platform setup for simultaneous synthesis of aryl diazonium libraries. a** Piping and instrument diagram and **b** photograph of a metal-based parallel flow platform setup for simultaneous synthesis of aryl diazonium libraries. The aryl diazonium reagent is injected through HPLC piston pump or diaphragm dosing Pump (**D1**, **D2**) into the reactor system, and distributed to 16 capillaries through the flow distributor. The building block reagents are introduced by syringe pumps (**I1–I8**) and digital peristaltic pumps (**I9–I16**). In addition, the Reglo Digital peristaltic pumps (**P1–P3**) are used to individually control the residence time of corresponding capillaries. **R1** is a capillary reactor composed of transparent PFA tubing for photoreaction, and **R2–R16** is composed of stainless-steel tubing.

clogging or in the presence of separate side feeds. Another consideration was the potential to decouple the flow of the main species stream from the flows of the building block species. All the modules and components were modelled using a 3D CAD (Computer-Aided Design) program to arrive at the design shown in Fig. 2a. The baffle discs with complex structures in the distributor were fabricated by 3D metal printing, and the components and modules were made by computer numerical control (CNC) machining. The manufactured modules were assembled by connecting to stainless-steel capillaries and pumps using union-type, T-type and bending-type Swagelok connectors, rendering a complete setup of metal-based parallel flow synthesizer as shown in Fig. 2b.

Figure 3a shows a schematic of the flow parallel synthesizer of Fig. 2a. Typically, the main species is introduced through both or one of the two main inlets (**D1** and **D2**) located at the bottom of the distributor, and cleaning liquid would be in-need introduced through the inlets. The distributor sends out equal amounts of the main species through *n* number of capillary feed lines, *n* being 16 in this case. Two coiled capillaries connected to both sides of the

T-mixer induce certain pressure drop as calculated in Supplementary Fig. 2, and it acts as a back pressure regulator to render the reliable flow distribution of main reagent at a few different flow rates of building blocks. The building blocks to react with the main species are introduced through 16 independent inlets (**I1** through **I16**) to each individual T-mixers (**T1** through **T16**) to be mixed with the main species. The mixed solutions proceed to individual capillary microreactors (**R1** through **R16**) for 16 independent reactions. These coiled capillary reactors are equipped with heating units for independent temperature control, as revealed by the IR imaging at two different temperatures, 75 and 100 °C (Supplementary Fig. 3). Peristaltic pumps (**P1–P3**) are installed for individual residence time adjustment in three capillaries. The system showed uniform flow distribution in the capillaries (**R4–R16**) regardless of the flow rate changes in the three capillaries (**R1–R3**). This is a phenomenon occurring due to the gravity predominantly acting on the system with a baffle-structure damper, creating a passively driven buffering effect[46,47]. The assembled actual flow parallel synthesizer, occupying an area of 35 cm × 35 cm, is shown in Fig. 3b.

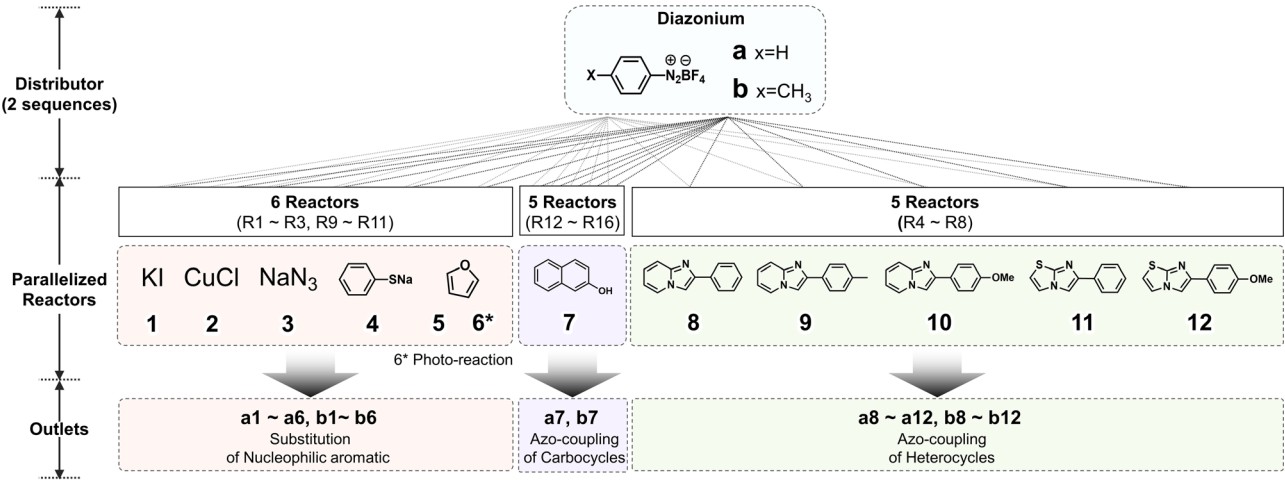

**Fig. 4 Schematic of aryl diazonium chemistries enabled by multiplex reactions of flow parallel synthesizer via 2 sequences of experiments.** A list of 12 building blocks are reacting with two types of diazonium salts to form six types of chemical bonds (two C–Halogen, C–N, C–S, C–C, –N=N–) in a library of 24 compounds.

To investigate the flow behaviour in the parallel synthesizer, computational fluid dynamics (CFD) analyses were conducted by utilizing the modelling results produced through the 3D CAD program (Supplementary Note 1, Supplementary Figs. 4, 6–8 and Supplementary Tables 1 and 2). The CFD results were also compared with the actual experimental flow rates (Supplementary Figs. 4–7 and Supplementary Tables 1 and 2). The uniformity of the flow distribution was quantified as a maldistribution factor (MF) that is a standard deviation of average mass flow rate at 16 capillaries. The numerical MF values at various flow rates are calculated to be less than 1%, while the experimental MF values measured by the collected amount of DMSO solvent as well as benzene diazonium tetrafluoroborate solution are less than 4% (Supplementary Figs. 4 and 5, Supplementary Table 1 and Supplementary Video 1). A low MF value corresponds to a more uniform flow distribution among capillaries[45].

The reservoir-type distributor with a baffle-structure damper provides some unique features. For one, it maintains uniform flow behaviour at somewhat higher rates, even when clogging occurs in a single or several capillaries, in the rest of the capillaries. This feature contrasts the conventional flow reactor systems embedded with bifurcation flow distributor that often causes a gradient flow profile along the remaining channels on clogging (Supplementary Fig. 6 and Supplementary Table 2)[45,48]. Another feature is that even when some of the capillaries are fed independently at different flow rates, uniform flow rates are maintained in the rest of the capillaries, which was confirmed both numerically and experimentally (Supplementary Fig. 7). These features allow multiplex synthesis under different conditions in reaction time and temperature. In addition, the dimensional effect of front coiled capillaries was thoroughly investigated by numerical analysis to compare decoupling of the main flow at all different flow rates of building blocks in the system (Supplementary Fig. 8). The longer and narrower coiled the capillary, the more clearly decouples the flow of the main species stream from the flow of the building block species.

**Flow parallel synthesis of aryl diazonium chemistry library and parameter screening.** Aryl diazonium forms a "transit hub" for arene chemistry from which almost any other aromatic derivative can be prepared, owes its chemical versatility to the $-N_2^+X^-$ functional group called "super electrophile" serving as a good leaving group. It is capable of reacting with any nucleophile such as hydrogen, oxygen, nitrogen, halogen, sulfur and carbon via ionic or radical pathways, to form almost any forms of bonds[49]. Moreover, the way of diverse functionalization can be an important model case to screen for preparation of novel chemical entities or generation of diverse chemical libraries that can serve either as leads in drug discovery or starting materials for next reactions[49]. Therefore, aryl diazonium-based reaction was chosen as a model for demonstrating a wide range of flow chemistries by performing efficient parameter screening and building synthesis library with the developed flow parallel synthesizer.

Initially, benzenediazonium tetrafluoroborate (**a**), one of the simplest and stable diazonium precursors, was used as the aryl diazonium salt to explore 6 aromatic substitution reactions, 1 azo-coupling reaction of a carbocycle at five different concentrations for the reasons elaborated shortly, and 5 azo-coupling reactions of heterocycles. All these 16 reactions took place simultaneously in the 16 microreactors, as shown in Fig. 4. After establishing optimal conditions, p-tolyldiazonium tetrafluoroborate (**b**) as alternative aryl diazonium salt was additionally used to generate product libraries.

In the experiment with benzenediazonium tetrafluoroborate (**a**), 0.77 M solution of **a** in DMSO was pumped using an HPLC pump through the two main inlets (**D1** and **D2**, Fig. 3) at six different flow rates ranging from 0.35 to 10.56 mL/min. This stream of diazonium solution was uniformly distributed into 16 individual stainless-steel capillaries of the flow synthesizer, the flow rate in the capillaries ranging from 0.022 to 0.66 ml/min, which corresponds to 1/16 of the total flow rate of the diazonium solution out of the distributor. The flow rate of the chemical building blocks entering the T-mixers was the same as that of diazonium stream.

The feed of chemical building block to the inlet **I1** that enters the mixer **M1** (refer to Fig. 3a) was a mixture of 7.7 M solution of furan and 5 mol% eosin Y (**6**) for a photochemical reaction. The feeds to the inlets **I2** through **I8** are: 0.77 M solution of CuCl (**2**) to **I2**, 0.77 M solution of KI (**1**) to **I3**, and 0.51 M solution of imidazopyridine (**8**, **9**, **10**) and imidazothiazole derivatives (**11**, **12**) to **I4**–**I8** for azo-coupling reactions of heterocycles. The feeds to the inlets **I9** through **I11** are: 0.77 M solution of $NaN_3$ (**3**) to **I9**, 0.77 M solution of sodium salt of p-thiocresol (**4**) to **I10** and

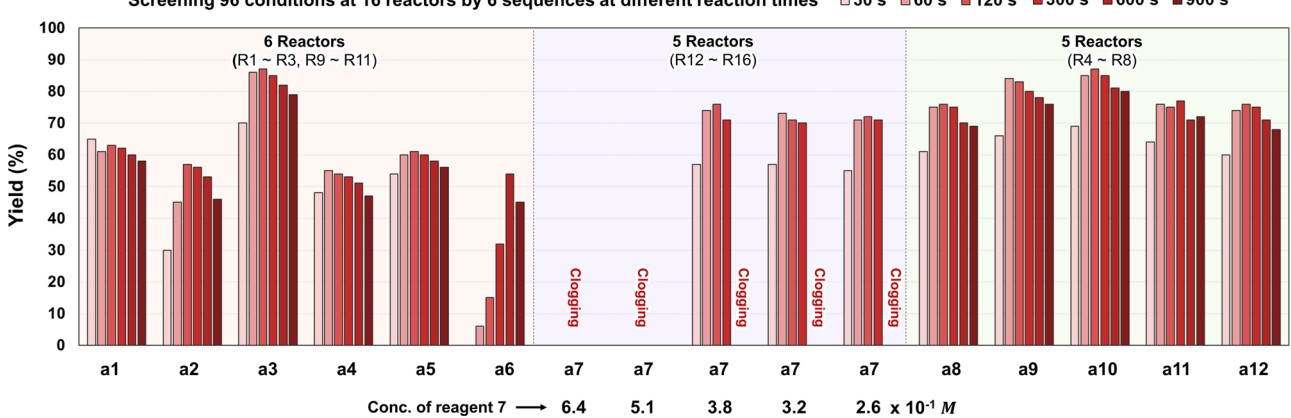

**Fig. 5 Screening 96 conditions at 16 reactors by 6 sequences at different reaction times.** Screening of reaction variables in reaction time (30–900 s) and concentration using a flow parallel synthesizer to find the optimal conditions of aryl diazonium-based chemical library. The red boxed region represents the syntheses based on substitution of nucleophilic aromatics, the purple boxed region represents azo-coupling of carbocycles and the green boxed region represents azo-coupling of heterocycles.

neat solution of furan containing 5 mol% of 4-aminomorpholine catalyst (**5**) to **I11** to demonstrate aromatic substitution-based reactions. Mixtures of 0.64, 0.51, 0.38, 0.32 and 0.26 M of $\beta$-naphthol (**7**) and NaOH were fed to the inlets **I12–I16**, respectively, for concentration screening of the azo dye-based carbocycle. The 16 chemical building block feeds mixed with the diazonium solution went through diverse bond forming reactions simultaneously in 16 capillary microreactors via substitution and radical pathways in a multiplex mode.

In the parallel synthesizer, 16 samples out of 16 microreactors (**R1–R16**) can be taken for the yield determination for a given residence time or flow rate once the experimental setup including pumps and feed solutions are ready. Six different residence times of 30, 60, 120, 300, 600 and 900 s were chosen for an optimization. At the end of screening 6 different residence times by merely changing the flow rate, $16 \times 6 = 96$ samples in total were collected to find the optimal parameters in reaction time and concentration that produced the highest yields, as summarized in Supplementary Table 3 and graphically shown in Fig. 5.

The 96 screening data tabulated in Supplementary Table 3 reveal some interesting facts that deserve comments. Notable is the fact that the optimal residence time for all the reactions except for 3 aromatic substitution-based reactions can be taken as 60 s, when a few percent higher yield is ignored in favour of lower residence time or higher throughput. An example is the reaction in the reactor **R6** for which the yield is 85% when the residence time is 60 s and it is 87% when the time is 120 s, for which 60 s was chosen as the optimum. These yields are comparable to those attained in batch reactors, some values being higher and some others lower than the batch yields. A sharp contrast in reaction time, however, is noted: 1 min for the continuous-flow reactor but anywhere from half an hour to 8 h for the batch reactor.

Of the 3 aromatic substitution-based reactions, the optimal residence time for the C–halogen bond formation of diazonium with KI (**1**) in capillary **R3** to give product (**a1**) was 30 s (65% yield) and that for the bond formation with CuCl (**2**) in capillary **R2** to give product (**a2**) was 120 s (57%). The lower yield of chlorobenzene was likely to be due to formation of small quantity of Gomberg–Bachmann product, a common side reaction generally encountered due to CuCl which results in the generation of aryl radical, responsible for the formation of *ortho/para*-chloro-1,1′-biphenyl. The optimal residence time for the photochemical C–C coupling reaction between diazonium and furan (**a6**) in capillary **R1** using green LED was 600 s (54% yield). Pre-decomposition of diazonium reagent due to extended

stay in the reservoir space of distributor led to generation of nitrogen gas, resulting in a yield lower than that of single capillary reactor (Supplementary Table 3). Fortunately, the unique design of our system allowed even distribution of the mixture of gas/solution, as shown in Supplementary Fig. 5. To accommodate the 3 different residence times for demonstrating the optimized reactions on a single flow platform, three peristaltic pumps **P1**, **P2** and **P3** (refer to Fig. 3a) were installed in the lines connected to the mixers **M1**, **M2** and **M3**. The diazonium flow rates corresponding to the residence times of 30, 120 and 600 s were 0.66, 0.17 and 0.033 mL/min, respectively, which will be explained shortly. Overall, six sets of screening ($16 \times 6 = 96$) were performed, involving syntheses based on aromatic substitution, azo-coupling of carbocycles and azo-coupling of heterocycles, to showcase the versatility of the flow parallel synthesizer.

One unique feature of the flow parallel synthesizer presented here is its ability to cope with clogging issues that can arise in any continuous-flow platform[42]. This system allows to continuously operate the non-clogged reactors even when certain reactors were blocked, without pausing the entire system. To demonstrate this capability, azo-dye synthesis was carried out in five reactors (**R12 – R16**) at five different concentrations (0.64, 0.51, 0.38, 0.32 and 0.26 M) for 6 different residence times (30, 60, 120, 300, 600 and 900 s), generating $6 \times 5 = 30$ concentration-based data points to arrive at an optimal concentration for good productivity. Clogging can be caused by precipitation of a poorly soluble product in DMSO solvents in azo-dye synthesis. As shown in Supplementary Table 3 and Supplementary Video 2, clogging occurred whatever the residence time is when the concentration was 0.51 M or higher. For lower concentrations of 0.26, 0.32 and 0.36 M, the clogging occurred only when the residence time was extended to 600 and 900 s. Long residence time or low flow rate often led to the deposition of dye product precipitates, causing the clogging. The optimal concentration chosen was 0.38 M for which the yield of Sudan dye was 74%. Whenever capillary clogging occurred, the total diazonium flow has to be reduced by 6.3% manually for each of the clogged capillaries and the corresponding building block flow halted. This procedure provided the same flow conditions for the rest of the capillaries and the reactions proceeded in the remaining reactors without interruption (Supplementary Fig. 6).

The total time required for screening $16 \times 6 = 96$ cases was approximately 60 min including the times needed to reach steady state for 6 different residence times, which is far less than the one performed in batch mode. The parallel synthesizer developed here

| Reactor number | R3 | R2 | R9 | R10 | R11, R1 | R12 ~ R16 | R4 ~ R6 | R7 ~ R8 |
|---|---|---|---|---|---|---|---|---|
| Building block | KI | CuCl | NaN$_3$ | (p-tolyl-SNa) | (furan) | (naphthol) | (2-arylimidazo[1,2-a]pyridine–X) | (2-arylthiazole–X) |
| Diazonium salts | **1**, 0.77 M | **2**, 0.77 M | **3**, 0.77 M | **4**, 0.77 M | **5**, neat [a] / **6**, 7.7 M [b] | **7**, 0.38 M | **8**, X = H, 0.51 M / **9**, X = CH$_3$, 0.51 M / **10**, X = OMe, 0.51 M | **11**, X = H, 0.51 M / **12**, X = OMe, 0.51 M |
| **Sequence 1** (N$_2$BF$_4$ benzene) **a**, 0.77 M | **a1**, 30 s, 65% | **a2**, 120 s, 57% | **a3**, 60 s, 85 % | **a4**, 60 s, 55% | **a5**, 60 s, 60% / **a6** [c], 60 s, 54% | **a7**, 60 s, 73% | **a8**, 60 s, 75% / **a9**, 60 s, 74% / **a10**, 60 s, 75% | **a11**, 60 s, 76% / **a12**, 60 s, 74% |
| **Sequence 2** (N$_2$BF$_4$ p-tolyl) **b**, 0.77 M | **b1**, 30 s, 69% | **b2**, 120 s, 66% | **b3**, 60 s, 81% | **b4**, 60 s, 64% | **b5**, 60 s, 62% / **b6** [c], 60 s, 61% | **b7**, 60 s, 78% | **b8**, 60 s, 77% / **b9**, 60 s, 73% / **b10**, 60 s, 75% | **b11**, 60 s, 76% / **b12**, 60 s, 71% |

**Fig. 6 Summary for optimized reaction conditions and yields for 24 parallel reactions.** All reactions were carried out at room temperature. The yield was calculated from isolated product. **a** 4-Aminomorpholine (5 mol%) was premixed with building block solution. **b** Eosin-Y (5 mol%) was premixed with building block solution. **c** 530 nm green LED was used. The red boxed region represents the syntheses based on substitution of nucleophilic aromatics, the purple boxed region represents azo-coupling of carbocycles, and the green boxed region represents azo-coupling of heterocycles.

is cost-effective, user friendly and highly efficient from screening point of view with minimum labour requirement.

Two different aryl diazonium salts, benzenediazonium tetrafluoroborate (**a**) and *p*-tolydiazonium tetrafluoroborate (**b**), were used to finally prove the multiplex synthesis of 24 compound libraries. The optimal conditions established for benzenediazonium tetrafluoroborate (**a**) were applied to *p*-tolydiazonium tetrafluoroborate (**b**) to generate 24 (12 × 2) product libraries in a time interval of 30 min, which included washing and stabilizing step to achieve steady state (Supplementary Video 3). The diazonium flow rates to **R1**, **R2** and **R3** were 0.033, 0.17 and 0.66 mL/min, respectively, and the flow rates to **R4–R16** were the same at 0.33 mL/min, bringing the total diazonium flow rate to 5.14 mL/min. The yields for all the samples collected from their respective outlets of the flow parallel synthesizer are summarized in Fig. 6. In general, the synthetic yield of the flow parallel synthesizer was comparable to those of batch and single capillary (Supplementary Note 2 and Supplementary Table 3). Scaling-up possibility with the flow parallel synthesizer was also demonstrated. The synthesis of Sudan dyes (**a7** and **b7**) using 5 capillary reactors (**R12–R16**) led to a throughput of 6.8 and 7.7 g/h, respectively, indicating that our parallel synthesizer can serve the dual role of screening and throughput scaling.

The main advantage of the flow parallel synthesizer developed here is the ability to perform multiplex screening with respect to substrates, concentrations, temperatures and residence times efficiently. These capabilities of the developed parallel synthesizer have not been fully provided by any commercially available batch parallel synthesizer or even automated flow platforms developed so far for high-end applications including efficient production of target molecules. Note that available platforms for reaction screening is comparatively tabulated (Supplementary Table 4).

## Conclusion

We have designed and developed the first flow parallel synthesizer that enables multiplex synthesis and optimization of compound libraries. Decoupling of all inlet flows to microreactors and design of the main flow distributor with a baffle structure have provided the parallel synthesizer with some unique capabilities. The synthesizer enables to perform its function without interruption upon clogging in flow lines, and allows independent control of the configured capillaries by adopting different reaction parameters in feeding rate, concentration and temperature. Multiple numbers, *n*, of reactions can be carried out simultaneously, bringing the number of cases to *n* × *m* when the multiplex synthesis is repeated *m* times in series in time. The desk-top size of miniaturized platform takes full advantage of the merits the microfluidics offer, leading to a short reaction time for the desired yield. With a model platform for which *n* is 16, aromatic substitution reactions (C–C, C–N, C–X and C–S bonds) and azo-coupling reactions for 96 different conditions were simultaneously screened for reaction variables, leading to optimal conditions in less than an hour. Based on this, multiplex synthesis of 12 × 2 compounds was demonstrated on a single flow platform. From academic and industrial perspective, this flow parallel synthesizer could minimize time, labour and capital investment, enhancing hit-to-lead optimization success ratio especially for pharmaceuticals from lab to commercialization. This system can be developed further for autonomous sequential multiplex synthesis by introducing artificial intelligence (AI) planning technology in the future, with features of automatic blockage detection and flow adjustment.

## Methods

**General methods**. All the reagents and solvents used were of commercial grade. All the reactions were carried out in a flow parallel synthesizer or single capillary reactor. Parts for the flow parallel synthesizer configuration were purchased from IDEX Health & Science LCC. The tube connecting the pump and platform consisted of high-purity PFA and PTFE tubes (1/16″ O.D., 0.75 mm I.D.) and polyether ether ketone 1/4 4–28 nuts. Swagelok tube fittings (SS-100-1-1, SS-600-1-2, SS-100-3 and SS-100-9) were purchased from Swagelok. Stainless-steel capillaries with 1/16″ O.D. and 0.75 mm I.D. were connected to the distributor body through the Swagelok connector. O-Rings (SM9-4D, heat resistant fluorine rubber, 8.5 pi O.D. and 1.5 mm thickness) were purchased from Misumi Korea. Reagents were

injected using a PHD Ultra syringe pump (Harvard Apparatus) equipped with an SGE glass syringes (SGE Analytical Science), constant flow gradient HPLC piston pumps (PrimeLine™ and Scientific Systems Inc.), SIMDOS diaphragm dosing Pump (KNF Group), digital peristaltic pump (ISMATEC) or Reglo Digital peristaltic pump (ISMATEC). The blockage in flow was simply detected by observing the stopped flow at the corresponding outlet, then physically cleared by feeding the DMSO solvent at a high flow rate (total flow rate of 10 ml/min per reactor) for several seconds. For the individual heating of the capillaries, the temperature system was configured as follows. A k-type thermocouple from Omega Engineering Korea was connected to NX2 proportional-integral-differential (PID)-based temperature controller from Hanyoung NUX to feed back the output value according to the current temperature of the capillaries reactor. The output value was converted to the high-voltage current entering cartridge-type rod heater from Super Heat Company, by Thyristor Power Regulator: WYU-DG 25 SI from Woonyoung Co., Ltd. All the reaction were monitored by thin layer chromatography (TLC) on Merck silica gel 60-F254-coated 0.25 mm plates, detected by UV, except for a few volatile compounds GC analysis was performed (Agilent 7890 A). Flash chromatography was performed with the indicated solvents on silica gel (particle size $0.064-0.210$ mm). Yields reported were made for the isolated, spectroscopically pure compounds. $^1H$ and $^{13}C$ NMR spectra were recorded on a Bruker-500 MHz and Bruker-300 MHz instrument with tetramethylsilane (TMS) as the internal standard. Chemical shifts are given in ppm ($\delta$), referenced to TMS ($\delta = 0.00$ ppm) or residual CHCl$_3$ peak ($\delta = 7.27$ ppm) for $^1H$ NMR, and CDCl$_3$ ($\delta = 77.0$ ppm) for $^{13}C$ NMR as internal standards. Data were represented as follows: chemical shift, multiplicity (s = singlet, d = doublet, t = triplet, m = multiples, b = broad, respectively), coupling constant (J, Hz) and integration (Supplementary Note 3, see Supplementary Information).

**Manufacturing and assembly of flow parallel synthesizer.** Metal 3D printing was performed using a Direct Metal Laser Sintering (DMLS) printer (ProX DMP320, 3DS Systems Inc.) with an accuracy of 50 µm. SUS630 17-4PH stainless-steel powder was used for 3D printing. In the case of CNC machining, machining of SUS316L was performed using CTX Beta 1250 TC equipment from DMG MORI. CNC machines provide position accuracy within 6 µm and repeatability within 2 µm. The individually manufactured parts were combined by placing polymer O-rings between the assembled parts to avoid leaks. The inlet/outlet tube and the stainless-steel capillaries were connected to the distributor body, the inlet or outlet junction through union-type, T-type and bending-type Swagelok connectors. The stainless-steel capillary (1/16″ O.D., 0.75 mm I.D.)-based microreactors were coiled for space-efficiency by bending the capillaries around a cylindrical spool (11 mm I.D, 1.6 mm pitch). In the fluid damper, the position of the baffle was set by the ratio of the height of the upper and the lower space by a ratio of 1:2, and the porosity value of baffle is 0.5 as optimized for the distribution performance[45].

**Computational fluid dynamics (CFD) simulation.** Fluid flow across the device, including distributors and capillaries, can be explained by the incompressible Navier–Stokes equations along with the mass conservation equation. Assuming steady state, the governing equation for fluid flow can be simplified as follows:

$$\rho v \cdot \nabla v = -\nabla p + \mu \nabla 2v + \rho g : \text{Navier} - \text{Stokes equation}$$

$$\nabla \cdot v = 0 : \text{Mass conservation equation}$$

$\rho$ is the fluid density, $v$ is the fluid linear velocity, $p$ is the pressure, $\mu$ is the fluid dynamic viscosity and $g$ is the gravitational acceleration. The governing equation was solved with proper boundary conditions that the outlet was set to atmospheric pressure in the general case. The corresponding flow condition was imposed in the case of the outlet forced by the peristaltic pump. Non-slip boundary conditions for speed and concentration were applied to the wall boundary. Because the Reynolds number of the present flows is less than 2000 (laminar flow), the steady incompressible Navier–Stokes equation was solved without using any turbulence model. The equations were discretized based on finite volume method and the commercialized numerical software COMSOL was used for the numerical simulation. The geometry of the channel was drawn by the INVENTOR (AUTODESK) preprocessor and structured meshes were generated with hexahedral elements with a typical size of 0.434 mm (in the case of circular cross-sectional channel, tetrahedral elements were used with a typical size of 0.434–0.885 mm). Typical number of meshes used for the simulation was around $10.48 \times 10^5$, which was confirmed by mesh size dependency check. The physical properties (density and viscosity) of DMSO at 25 °C (1100 kg/m$^3$, 1.987 mPa·s), which was the carrier solvent, were used for fluid properties. Iteration numbers for convergence were from 10 times in the minimum to over 1000 times in the maximum.

**Computational and experimental fluid dynamics of flow parallel synthesizer.** The flow performance of the parallel synthesizer was evaluated by numerical analysis of CFD and preliminary flow experiment of the manufactured system. The uniformity of the flow distribution was quantified using the maldistribution factor

(MF)[45,48,50,51].

$$\text{MF}(\%) = \sqrt{\frac{1}{n-1} \sum_{i=1}^{n} \left(\frac{m_i - \bar{m}}{\bar{m}}\right)^2} \times 100$$

$n$ is the number of capillaries, $m_i$ is the mass flow rate of the $i$th capillary. $\bar{m}_i$ represents the average mass flow rate at each capillary. Consequently, the MF value is the standard deviation of the mass flow rate in each capillary. Accordingly, a low MF value indicates a uniform flow distribution between capillaries. The clogging of the capillary was implemented by considering specific outlet as a wall in the CFD program.

**Solution preparation of diazonium salts and other building blocks**
*Diazonium solution.* 100 mL, 0.77 M solution of respective diazonium salts (77 mmol) were prepared using DMSO.

Building Block 1: 10 mL, 0.77 M solution of KI (7.7 mmol) were prepared using DMSO:H$_2$O (9:1).

Building Block 2: 10 mL, 0.77 M solution of CuCl (7.7 mmol) were prepared using DMSO:HCl (3:2).

Building Block 3: 10 mL, 0.77 M solution of NaN$_3$ (7.7 mmol) were prepared using DMSO:H$_2$O (9:1).

Building Block 4: 10 mL, 0.77 M solution of *p*-thiocresol and NaOH (7.7 mmol each) were prepared using DMSO:H$_2$O (2.5:1).

Building Block 5: 10 mL of neat furan solution containing 4-aminomorpholine (0.385 mmol, 5 mol%).

Building Block 6: 10 mL, 7.7 M solution of furan (77 mmol) and eosin Y (0.385 mmol, 5 mol%) were prepared using DMSO.

Building Block 7: 10 mL, each 0.6375, 0.51, 0.3825, 0.31875 and 0.255 M solution of β-naphthol and NaOH (6.375, 5.1, 3.825, 3.1875 and 2.55 mmol each) were prepared using DMSO:H$_2$O (9:1).

Building Block 8–12: 5 mL, 0.51 M solution of imidazopyridine/imidazothiazole (2.55 mmol) were prepared using DMSO.

## Data availability
The data that support the findings of this study are available from the corresponding author upon reasonable request.

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

## Acknowledgements

This work was supported by the National Research Foundation of Korea (NRF) grant funded by the Korea government (MSIP) (NRF-2017R1A3B1023598).

## Author contributions

D.K., G.A. and B.S. designed research; G.A. designed and manufactured the synthesizer; G.A. and B.S. performed experiments and analysed the data; S.L. contributed in initial single capillary experiments, prepared sample and analysed the data; S.V. helped to analyse the data; S.Y. performed CFD simulation; G.A., B.S. and D.K. wrote the paper.

## Competing interests

The authors declare no competing interests.
