## [Peer Review File · Communications Chemistry]

Reviewers' comments:

Reviewer #1 (Remarks to the Author):

The MS is very interesting and will promote good discussion but there are some points which need addressing first.

The sign-posting of information is not very clear and could be strengthened to highlight the great work the group have done. I have detailed points below in three categories, clogging, general and CFD.

Regarding the clogging:

1. The precise manner of clogging is not defined (viscous reaction mixture / solids / etc) nor is any indication of how this was cleared after blockage. As the MS rightly points out, clogging is a major problem and how to clear a blockage is an important part of reactor use, therefore I think the blockage clearing method used would be very useful for anyone reading this paper.
2. In lines 226-227, it states that the reactor is able to "handle clogging issues". This statement is a little misleading as it suggests to me that it would prevent clogging rather than be able to continue to run parallel synthesis in the non-clogged reactors.
3. The manner in which clogging was detected has not been given. This is important for the presentation of this kind of reactor.
4. Lines 237-238 state that the flow rate of the main flow was reduced when clogging occurred. This should be signposted sooner as it is an extremely important and very good point of the reactor. The data presented in Figure S4 do not appear to have this adjustment. If there was adjustment the flow rates should be around 0.64 ml/min, the graph shows most of the values around 0.68 ml/min.
5. In Figure S4, the manner of clogging of the reactor was not presented. Was this intentional or during a synthesis run?
6. How repeatable were the clogging events? In Table S3, how many trials were carried out for each data set presented?

General:

7. The variation of flow rate shown in Figure S4 appear to be significantly above the presented MF values.
8. Are there any data (experimental or CFD) for the efficiency of the 5 mm ID mixing chamber for D1 and D2 (looking at different flow rates).
9. I would be interested if the authors looked at different (unequal) secondary flow rates (from I1-I16, Fig 3a) and whether there was a change in the flow rate coming through from the main flow (D1/2) to those capillaries with e.g. a reduced secondary flow rate (thus perhaps reduced pressure at mixer pieces).
10. The thruhole size for the mixer t-piece used as a secondary inlet was not mentioned. This is needed as much as the volume (which is given).
11. In Figure S1b, the terms need better definition. Some are explicit but e.g. Vjunction is not clear as I would have guessed this as the same as Vmixer. In Figure S4 caption, 'MF' needs defined.
12. The pressure of the system has not been given for any point of the reactor. A back-pressure regulator (BPR) has not been mentioned in the design / methodology, this would be essential for this type of reactor. Please specify exactly what BPR is used.
13. The syringe and HPLC pumps used have been detailed but the peristaltic pump(s) have not. In Figure 3b caption, HPLC pumps are not mentioned. I presume this is the 'continuous pumps', this term can refer to many pump types including peristaltic.
14. In line 212, R1-4 must be transparent, this seems evident in Figure 3b, but this has not been detailed anywhere in the text, so far all reactors are presented as stainless steel. Please give exact details of transparent reactors.

CFD:

15. There are not enough details for the CFD. Whilst very basic CFD information is given, however,

e.g. mesh analysis, iteration numbers and model details are missing. How clogging was simulated is also not given.

Reviewer #2 (Remarks to the Author):

The authors have developed a parallel flow synthesizer consisting of the unique built-in distributor and 16 capillary-type micro reactors, and applied to multiplex synthesis and rapid optimization of chemical libraries. It is noteworthy that the flow distributors with the baffle discs allowed the uniform flow distribution even when clogging occur. This excellent system not only enables the rapid optimization of multiple library synthesis in a short period, but also allows to easily scaling-up by the numbering up approach.

The paper provides very interesting strategy and results but it still needs a considerable revision to be acceptable for communication chemistry.

<Major comments>

#1

Maintaining a uniform flow distribution of the main feed solution (D1 or D2) even when performing parallel synthesis with multiple reagent solutions (I1~16), is most important to demonstrate this strategy. When various type of building block solutions that would have different physical properties (viscosity, etc.) were introduced into the microreactor, each microreactor would exhibit different pressure drops. In that case, it is easily expected that maintaining a uniform flow distribution of the main feed flow into each microreactors will be more difficult. If the uniformity is broken, the proposed strategy will not work well.

So, the authors should confirm and reveal the relationship between the pressure drop at capillary type microreactors and the uniformity of flow distribution. In this paper, the authors does not show evidence about the uniformity of the main feed during parallel synthesis using various reagents. The data in Figures S2, S3, S4 and Table S1, seems to be the results when the same solvent or solution was used at I1~16.

#2

Line 89: One main design consideration was to decouple the flow of the main species stream from the flows of the building block species such that any malfunctioning such as clogging in any feed line does not affect the rest of the lines.

It seems that the main species flow and the building block flow are not decouple. What dose decouple mean? I this this part needs more explanation.

<Minor comments>

#3

Line 130: For one, it maintains uniform flow behavior at somewhat higher rates, even when clogging occurs in a single or several capillaries, in the rest of the capillaries.

This result is excellent. However, if the flow rate of other capillaries increases due to the clogging, the molar ratio of the diazonium salt and reagents will be changed. Does this change affect the desired reaction? In addition, this result is inconsistent with the claim on line 89.

#4

Line 237: Whenever capillary clogging occurred, the total diazonium flow was reduced by 6.3% for

each of the clogged capillaries and the corresponding building block flow halted. This procedure provided the same flow conditions for the rest of the capillaries and the reactions proceeded in the remaining reactors without interruption.

Does this mean that if clogging occurs, we can manually change the flow rate of main species feed?

It would be better if it was automatically changed when a clogging was detected.

#5

Table 3 in SI

What does capillary yield in Table S3 mean? It is the result when the desired reaction is performed by single capillary type microreactor with the optimized condition?

If so, the authors should add a description.

Reviewer #3 (Remarks to the Author):

The paper of Kim et al. is outstanding both in terms of its conceptual depth and innovation as well as breath of demonstrated synthetic innovation.

It is so true what the authors mention. Microreactors and flow chemistry are so successful in chemical laboratory development and even in pharmaceutical production. Yet their initial destination was also to speed up drug discovery, done at the background of combinatorial synthesis, ranked so high about 20 years ago. GSK and others have investigated this. Yet the "discovery promise" has never been kept. The Kim paper sets a very new and original tone here.

I am impressed by the depth of the systems approach and its degree of actual realisation and utilisation. Chemistry and process engineering aspects are intertwined to their mutual advantage.

I firmly endorse publication of this paper, with suggesting minor revisions, as follows.

1) While it is good to coin the low degree of "discovery-microreactor" achievement by the early "2 x 2"-paper, a more comprehensive state of the art summary should be given. The formation of somewhat larger "Y x Y" libraries in microreactors has been reported.

2) The authors should state why those early concepts failed. The leading investigators were excellent people. So that cannot be the point. What is the new key knowledge that enables the authors to perform better than their colleagues?

3) The elegant serial screening approach of de Bellefon et al. using multiphase flow is not mentioned (Angew Chem, Adv Synth Cat; published about 20 years ago). It should. de Bellefon benchmarked nicely and showed - quantitatively - how flow can improve screening as compared to batch. The authors should make similar comparison.

4) A quick summary of flow distribution approaches would be helpful. This is central to the Kim-concept. Especially, the Hasebe and Commenge works are much relevant. Again, it would be nice to see why on the background of those good concepts the Kim concepts were chosen.

5) A comparison to modern HTS performance would be good (as a table). What does industry's state of the art deliver and what might the Kim concept be able to do under optimistic assumptions? It is one thing to be better than 20 years old concepts and another to make a valuable contribution to today's industry.

6) I think the authors would agree that use of computing and artificial intelligence is so relevant

for such screening. They may add a comment on this; e.g. as an Outlook.

7) Another good topic for an Outlook would be if the concept is amenable to multiphase reactions, which are important for medicinal synthesis. And how to achieve that. Flow distribution is (much) more difficult. A critical view would be welcome here.

Dear reviewers,

Authors greatly appreciate reviewers for the valuable comments and suggestions that helped us to improve substantially the quality of the manuscript. According to the suggestions and requests, and comments of the reviewers, authors have carefully revised the manuscript as highlighted in yellow. A separate response to reviewers is also prepared as the below. In particular, to answer the technical queries from reviewers, we have performed additional numerical analysis. Authors believe that these modifications have strengthened the manuscript and hope that all reviewer concerns have been suitably addressed in the manuscript, and the revised manuscript now meets the standard for publication in *Communications Chemistry*.

Thank you very much for your attention and consideration.

Sincerely,

Dong-Pyo Kim
Center of Intelligent Microprocess for Pharmaceutical Synthesis,
Department of Chemical Engineering,
Pohang University of Science and Technology (POSTECH),
Pohang, South Korea

E-mail: dpkim@postech.ac.kr

Reviewer #1:

The MS is very interesting and will promote good discussion but there are some points which need addressing first.

The sign-posting of information is not very clear and could be strengthened to highlight the great work the group have done. I have detailed points below in three categories, clogging, general and CFD.

Regarding the clogging:

1. The precise manner of clogging is not defined (viscous reaction mixture / solids / etc) nor is any indication of how this was cleared after blockage. As the MS rightly points out, clogging is a major problem and how to clear a blockage is an important part of reactor use, therefore I think the blockage clearing method used would be very useful for anyone reading this paper.

3. The manner in which clogging was detected has not been given. This is important for the presentation of this kind of reactor.

5. In Figure S4, the manner of clogging of the reactor was not presented. Was this intentional or during a synthesis run?

6. How repeatable were the clogging events? In Table S3, how many trials were carried out for each data set presented?

Answer of 1,3, 5 and 6: In the parallel reactor, two different cases of clogging occurred. One is to intentionally block the specific outlet flow by installing a cap at the outlet of the T-mixer (see **Supplementary Fig. 6**) in the simply repeated manner, the other is to experimentally block by precipitation of poorly soluble products from azo-dye synthesis in DMSO solvent, in particular, at high concentrations or low flow rates (see **Supplementary Table 3 Entry # 37-66**). At the latter cases, authors clarified by newly adding the blockage detection and clearing method to the **Methods** section (**page 14**): “The blockage in flow was simply detected by observing the stopped flow at the corresponding outlet, then physically cleared by feeding the DMSO solvent at a high flow rate (total flow rate of 10 ml/min per reactor) for several seconds.” The experimental results in **Supplementary Fig. 6** are the average values and deviations of the results of repeated experiments more than 3 times. The data presented in **Supplementary Table 3**, including observations of clogging phenomenon by azo-dye synthesis, is a summary of the results performed twice, and the occurrence of clogging in each condition was reproducibly observed.

2. In lines 226-227, it states that the reactor is able to “handle clogging issues”. This statement is a little misleading as it suggests to me that it would prevent clogging rather than be able to continue to run parallel synthesis in the non-clogged reactors.

Answer: The sentence was revised to “One unique feature of the flow parallel synthesizer presented here is its ability to **cope with** clogging issues that can arise in any continuous flow platform. **This**

system allows to continuously operate the non-clogged reactors even when certain reactors were blocked, without pausing the entire system.” from “One unique feature of the flow parallel synthesizer presented here is its ability to handle clogging issues that can arise in any continuous flow platform.” at **Results and Discussion** section (page 11).

4. Lines 237-238 state that the flow rate of the main flow was reduced when clogging occurred. This should be signposted sooner as it is an extremely important and very good point of the reactor. The data presented in Figure S4 do not appear to have this adjustment. If there was adjustment the flow rates should be around 0.64 ml/min, the graph shows most of the values around 0.68 ml/min.

Answer: For better understanding, authors newly added a table into **Supplementary Fig. 6** to show how to adjust the total flow rate (**D1+D2**) for maintaining the initial flow (0.66 ml/min) of the non-clogged reactors at various cases. Moreover, this unique feature of this system was strongly described in detail at the “Design Principle of Flow Parallel Synthesizer” of **Results and Discussion** section (page 6-7).

General:

7. The variation of flow rate shown in Figure S4 appear to be significantly above the presented MF values.

Answer: It was revised in **Supplementary Table 2** as the following “MF values obtained from **Supplementary Fig. 6**, calculated excluding the clogged channels.”.

8. Are there any data (experimental or CFD) for the efficiency of the 5 mm ID mixing chamber for D1 and D2 (looking at different flow rates).

Answer: In this work, **D1 & D2** were simply used for sequential injection of reactants and washing solvent with no need of mixing. In the preceding study (**ref. 45** : Study on the numbering-up metal microreactor for high-throughput production of drugs), a numerical analysis study on the mixing efficiency was performed by installing the static mixer into the 5 mm ID mixing chamber.

9. I would be interested if the authors looked at different (unequal) secondary flow rates (from I1-I16, Fig 3a) and whether there was a change in the flow rate coming through from the main flow (D1/2) to those capillaries with e.g. a reduced secondary flow rate (thus perhaps reduced pressure at mixer pieces).

12. The pressure of the system has not been given for any point of the reactor. A back-pressure regulator (BPR) has not been mentioned in the design / methodology, this would be essential for this type of reactor. Please specify exactly what BPR is used.

Answer of 9 and 12: It is important for practical use that the main flow can be decoupled from the building block flow. In this work, three peristaltic pumps (**P1 to P3**) were installed to adjust individual flow rate of three capillary reactors (**R1~R3**) in a decoupling manner from the flow rate of the other

capillaries (**R4~R16**) in the system. And it was discussed that the gravity predominantly acted on the system with the baffle-structure damper, creating a passively driven buffering effect (see **page 6** and **Fig. 3a.**). In addition, as newly described at **Results and Discussion** section in the revised manuscript (**page 6**), “Two coiled capillaries connected to both sides of T-mixer induce certain pressure drop as calculated in **Supplementary Fig. 2**, and it acts as a back pressure regulator to render the reliable flow distribution of main reagent at a few different flow rates of building blocks.” “In addition, the dimensional effect of front coiled capillaries was thoroughly investigated by numerical analysis to compare decoupling of the main flow at all different flow rates of building blocks in the system (**Supplementary Fig. 8**). The longer and narrower coiled capillary, the more clearly decouples the flow of the main species stream from the flow of the building block species.” which was added to the **Results and Discussion** section (**page 7**) of the revised manuscript.

10. The thruhole size for the mixer t-piece used as a secondary inlet was not mentioned. This is needed as much as the volume (which is given).

Answer: Dimension of T-mixer was added to **Supplementary Fig. 1**.

11. In Figure S1b, the terms need better definition. Some are explicit but e.g. V_{junction} is not clear as I would have guessed this as the same as V_{mixer} . In Figure S4 caption, ‘MF’ needs defined.

Answer: In **Supplementary Fig. 1**, the detailed information including diameter of T-mixer and L-fitting was newly added, and some expressions that were confusing have been also corrected. The definition of MF was specified in “Computational and experimental fluid dynamics of flow parallel synthesizer” of **Methods** section in the manuscript (**page 16**).

13. The syringe and HPLC pumps used have been detailed but the peristaltic pump(s) have not. In Figure 3b caption, HPLC pumps are not mentioned. I presume this is the ‘continuous pumps’, this term can refer to many pump types including peristaltic.

Answer: Various continuous pumps such as HPLC piston pumps, diaphragm infusion pump and digital peristaltic pumps were used, and the detailed was newly described at **Fig. 3** and the **Methods** section of the revised manuscript (**page 14**).

14. In line 212, R1 must be transparent, this seems evident in Figure 3b, but this has not been detailed anywhere in the text, so far all reactors are presented as stainless steel. Please give exact details of transparent reactors.

Answer: Transparent PFA tubing (IDEX Health & Science LCC., I.D. 0.75 mm, O.D. 1.58 mm) was used for photoreaction. The detailed was newly added into **Fig. 3**, with tubing size in the **Methods** section (**14 page**): “**R1** is a capillary reactor composed of transparent PFA tubing for photoreaction, and **R2~R16** is composed of stainless steel tubing.”, “The tube connecting the pump and platform

consisted of high purity PFA and PTFE tubes (1/16" O.D., 0.75 mm I.D.) and polyether ether ketone 1/4 4-28 nuts.”.

CFD:

15. There are not enough details for the CFD. Whilst very basic CFD information is given, however, e.g. mesh analysis, iteration numbers and model details are missing. How clogging was simulated is also not given.

Answer: The detailed information of CFD analysis was newly added to the **Methods** section at **pages 15-16** of the revised manuscript. The clogging of the capillary was implemented by regarding a specific outlet as a wall in the CFD program.

Reviewer #2:

The authors have developed a parallel flow synthesizer consisting of the unique built-in distributor and 16 capillary-type micro reactors, and applied to multiplex synthesis and rapid optimization of chemical libraries. It is noteworthy that the flow distributors with the baffle discs allowed the uniform flow distribution even when clogging occur. This excellent system not only enables the rapid optimization of multiple library synthesis in a short period, but also allows to easily scaling-up by the numbering up approach.

The paper provides very interesting strategy and results but it still needs a considerable revision to be acceptable for communication chemistry.

Major comments

1. Maintaining a uniform flow distribution of the main feed solution (D1 or D2) even when performing parallel synthesis with multiple reagent solutions (I1~16), is most important to demonstrate this strategy. When various type of building block solutions that would have different physical properties (viscosity, etc.) were introduced into the microreactor, each microreactor would exhibit different pressure drops. In that case, it is easily expected that maintaining a uniform flow distribution of the main feed flow into each microreactors will be more difficult. If the uniformity is broken, the proposed strategy will not work well.

So, the authors should confirm and reveal the relationship between the pressure drop at capillary type microreactors and the uniformity of flow distribution. In this paper, the authors does not show evidence about the uniformity of the main feed during parallel synthesis using various reagents. The data in Figures S2, S3, S4 and Table S1, seems to be the results when the same solvent or solution was used at I1~16.

2. Line 89: One main design consideration was to decouple the flow of the main species stream from the flows of the building block species such that any malfunctioning such as clogging in any feed line does not affect the rest of the lines.

It seems that the main species flow and the building block flow are not decouple. What dose decouple mean? I this this part needs more explanation.

Answer of 1 and 2: It is important for practical use that the main flow can be decoupled from the building block flow. In this work, three peristaltic pumps (**P1 to P3**) were installed to adjust individual flow rate of three capillary reactors (**R1~R3**) in a decoupling manner from the flow rate of the other capillaries (**R4~R16**) in the system. And it was discussed that the gravity predominantly acted on the system with the baffle-structure damper, creating a passively driven buffering effect (see **page 7** and **Fig. 3a.**). In addition, as newly described at **Results and Discussion** section in the revised manuscript (**page 6**), “Two coiled capillaries connected to both sides of T-mixer induce certain pressure drop as calculated in **Supplementary Fig. 2**, and it acts as a back pressure regulator to render the reliable flow distribution of main reagent at a few different flow rates of building blocks.” “In addition, the dimensional effect of front coiled capillaries was thoroughly investigated by numerical analysis to compare decoupling of the main flow at all different flow rates of building blocks in the system (**Supplementary Fig. 8**). The longer and narrower coiled capillary, the more clearly decouples the flow of the main species stream from the flow of the building block species.” which was added to the **Results and Discussion** section (**page 7**) of the revised manuscript.

Minor comments

3. Line 130: For one, it maintains uniform flow behavior at somewhat higher rates, even when clogging occurs in a single or several capillaries, in the rest of the capillaries.

This result is excellent. However, if the flow rate of other capillaries increases due to the clogging, the molar ratio of the diazonium salt and reagents will be changed. Does this change affect the desired reaction? In addition, this result is inconsistent with the claim on line 89.

Answer: Authors newly added a table into **Supplementary Fig. 6** to show how to adjust the total flow rate (**D1+D2**) for maintaining the initial flow condition under various clogging cases. In specific, it was described at **page 11-12**: “Whenever capillary clogging occurred, the total diazonium flow has to be reduced by 6.3% manually for each of the clogged capillaries and the corresponding building block flow halted.”. Eventually, this system allows to continuously operate the non-clogged reactors even when certain reactors were blocked, without pausing the entire system. Actually, it was demonstrated by the case of precipitation block from azo-dye synthesis in DMSO solvent (see **Supplementary Table 3 Entry # 37-66**).

4. Line 237: Whenever capillary clogging occurred, the total diazonium flow was reduced by 6.3% for each of the clogged capillaries and the corresponding building block flow halted. This procedure provided the same flow conditions for the rest of the capillaries and the reactions proceeded in the remaining reactors without interruption.

Does this mean that if clogging occurs, we can manually change the flow rate of main species feed?

It would be better if it was automatically changed when a clogging was detected.

Answer: Currently, the flow rate of main reagent was manually adjust when the blockage was detected by observing the stopped flow at the corresponding outlet. Authors clarified by newly adding the blockage detection method to the **Methods** section (**page 14**): “The blockage in flow was simply detected by observing the stopped flow at the corresponding outlet, then physically cleared by feeding the DMSO solvent at a high flow rate (total flow rate of 10 ml/min per reactor) for several seconds.” As suggested, authors are undergoing to develop an automation system to cope with adjustment of flow conditions in a computer-operated manner, and added the perspective point to the **Conclusion**

section (page 14).

5. Table 3 in SI. What does capillary yield in Table S3 mean? It is the result when the desired reaction is performed by single capillary type microreactor with the optimized condition?

If so, the authors should add a description.

Answer: Yes, the capillary yield in **Supplementary Table 3** under optimized conditions was made to confirm the performance of the flow parallel synthesizer, as already mentioned (see **2nd paragraph of page 12**). For details such as reactor dimensions, please refer to the **page 16 of supplementary information**.

Reviewer #3:

The paper of Kim et al. is outstanding both in terms of its conceptual depth and innovation as well as breath of demonstrated synthetic innovation.

It is so true what the authors mention. Microreactors and flow chemistry are so successful in chemical laboratory development and even in pharmaceutical production. Yet their initial destination was also to speed up drug discovery, done at the background of combinatorial synthesis, ranked so high about 20 years go. GSK and others have investigated this. Yet the "discovery promise" has never been kept. The Kim paper sets a very new and original tone here.

I am impressed by the depth of the systems approach and its degree of actual realisation and utilisation. Chemistry and process engineering aspects are intertwined to their mutual advantage.

I firmly endorse publication of this paper, with suggesting minor revisions, as follows.

1. While it is good to coin the low degree of "discovery-microreactor" achievement by the early "2 x 2"-paper, a more comprehensive state of the art summary should be given. The formation of somewhat larger "Y x Y" libraries in microreactors has been reported.

Answer: It was newly discussed in **Introduction** section (page 2) of the revised manuscript as below, "Formation of the "A x B" libraries in flow are mostly based on segmented flow system with highly enhanced mass and heat transfer. leading to screen catalysts (ref. 33 - 35), combinatorial chemistry (ref. 12 and 17), and the nanomaterials (ref. 36). However, these approaches have limitations in direct utilization of the derived optimal reaction conditions into continuous-flow process." Therefore, it demands a new efficient platform to speed up discovery of diverse chemistry in a continuous-flow manner. This work enables to perform simultaneous multiplex synthesis in a continuous-flow type of parallel synthesizer under diverse reaction conditions, instead of a low throughput single reactor. Moreover, the derived reaction conditions could be directly realized and utilized for sequential library synthesis and practical scale-up process. These aspects have been added in the revised manuscript in **Introduction** section (page 2-3).

2. The authors should state why those early concepts failed. The leading investigators were excellent people. So that cannot be the point. What is the new key knowledge that enables the authors to

perform better than their colleagues?

Answer: In essence, we are providing a different perspective and concept of parallelization, which here means simultaneous execution of a number of different reactions, beyond traditionally meant increasing throughput. This aspect has been added on **page 4** of the revised manuscript. The key technology related to uniform flow distribution is discussed below in our response to **the comments of question 4**.

3. The elegant serial screening approach of de Bellefon et al. using multiphase flow is not mentioned (Angew Chem, Adv Synth Cat; published about 20 years ago). It should. de Bellefon benchmarked nicely and showed - quantitatively - how flow can improve screening as compared to batch. The authors should make similar comparison.

Answer: De Bellefon and colleagues' segment flow screening method was one of the beginning of all-segment flow-based automated screening system. Relevant technical summary up to date and differentiation of this study have been added to **Introduction** section (**page 2**).

4. A quick summary of flow distribution approaches would be helpful. This is central to the Kim-concept. Especially, the Hasebe and Commenge works are much relevant. Again, it would be nice to see why on the background of those good concepts the Kim concepts were chosen.

Answer: Various types of parallel microreactor for the purpose of increasing throughput have been attempted by many researchers, including numerical analysis and industrial purpose (**ref. 38 - 40**). Alternatively, Hasebe et al. have not only optimized parameters for parallelization of the manifold (**ref. 41**), but also developed efficient clogging detection using minimal sensors (**ref. 42**), and demonstrated in actual process control(**ref. 43 and 44**). Here, we present a metal-based flow parallel synthesizer for synthetic screening and optimization. This flow parallel synthesizer has a damper-based distributor with baffles that enables uniform flow distribution and allows the system to operate without interruption when clogging occurs in some of the capillaries. Therefore, the synthesizer can concurrently execute multiple reactions in parallel under diverse reaction conditions, including photochemistry. This discussion has been added on **page 4** of the revision.

5. A comparison to modern HTS performance would be good (as a table). What does industry's state of the art deliver and what might the Kim concept be able to do under optimistic assumptions? It is one thing to better than 20 years old concepts and another to make a valuable contribution to today's industry.

Answer: The comparison of this study with the existing platforms for HTS has been made in the revised Supplementary Information, **Supplementary Table 4**. Our concept makes it possible to carry out a variety of individual reactions simultaneously, on individual capillaries connected in parallel using the reliable flow distributor. This new platform enables efficient screening of optimized conditions that is compatibly employed to practical scale-up systems.

6. I think the authors would agree that use of computing and artificial intelligence is so relevant for such screening. They may add a comment on this; e.g. as an Outlook.

Answer: The following comment has been added in the last sentence of the **Conclusions** section on **page 14** of the revised manuscript. "This system can be developed further for autonomous sequential multiplex synthesis by introducing artificial intelligence (AI) planning technology in the future, with features of automatic blockage detection and flow adjustment."

7. Another good topic for an Outlook would be if the concept is amenable to multiphase reactions, which are important for medicinal synthesis. And how to achieve that. Flow distribution is (much) more difficult. A critical view would be welcome here.

Answer: It may be possible to implement an emulsification process or a nanomaterial synthesis by transforming the system configuration for making large volumes of droplets uniformly containing diverse chemical combination, as a new practical tool of nano-medicine research.

REVIEWERS' COMMENTS:

Reviewer #1 (Remarks to the Author):

I am happy with all additions and amendments and believe the paper should be published in its current state.

Reviewer #2 (Remarks to the Author):

The presented manuscript has been revised adequately. The additional data and commentary helped resolve my concerns. In particular, the result shown in Figures S2 and S8 is an important finding that enhances the value and reliability of the method proposed in this paper. I appreciate the authors for their collaboration.

I recommend that it be accepted for publication in communications chemistry.